# On Some Fixed Point Iterative Schemes with Strong Convergence and Their Applications

Anku [1], Mona Narang [2] and Vinay Kanwar [3,*]

1    Department of Mathematics, Panjab University, Chandigarh 160014, India; ankusuthar45@gmail.com
2    Department of Mathematics, D.A.V. College, Chandigarh 160010, India; monanarang@yahoo.com
3    Department of Mathematics, University Institute of Engineering and Technology, Panjab University, Chandigarh 160014, India
\*    Correspondence: vmithil@yahoo.co.in

**Abstract:** In this paper, a new one-parameter class of fixed point iterative method is proposed to approximate the fixed points of contractive type mappings. The presence of an arbitrary parameter in the proposed family increases its interval of convergence. Further, we also propose new two-step and three-step fixed point iterative schemes. We also discuss the stability, strong convergence and fastness of the proposed methods. Furthermore, numerical experiments are performed to check the applicability of the new methods, and these have been compared with well-known similar existing methods in the literature.

**Keywords:** fixed point method; nonlinear equation; order of convergence; strong convergence





## 1. Introduction

The theory of fixed point iterative methods has progressively become an invaluable area of study as many problems in mathematics, engineering, physics, economics etc. can be transformed into a fixed point problems [1,2]. In fact, variational inequalities and equilibrium problems in Hilbert spaces and Banach spaces are solved by using fixed point iterative schemes [3–6]. These techniques have been applied in fluid mechanics and fluid–structure interaction [7,8], the design and analysis of fractals, etc. Over the years, researchers have developed several iterative methods for the approximation of fixed points [9]. Fixed point iterative schemes are eminent as every root finding problem can also be converted into a fixed point problem and vice versa. Suppose we want to find the solution of a nonlinear equation

$$f(x) = 0, \tag{1}$$

where $f : [a, b] \subset \mathbb{R} \to \mathbb{R}$ is sufficiently differentiable function with simple zeros. This equation can also be written in the form

$$x = T(x), \tag{2}$$

such that any solution of Equation (2), which is fixed point, is a root of the Equation (1).

Let $(E, ||.||)$ be a real Banach space and $T : D \to D$ be a function, where $D$ is a non-empty closed and convex subset of $E$ and $x_0 \in D$. Let $F_T = \{r \in D | Tr = r\}$ represent the set of all fixed points of $T$. One of the oldest, most simple and best-known fixed point iterative methods for approximating the fixed points was given by Picard [10] in 1890, which is given by

$$x_{j+1} = T(x_j), \ j \geq 0. \tag{3}$$

Picard iteration approximates fixed points of Equation (2), where $T$ is contraction mapping. If $T$ is non-expansive, then the Picard iterative process fails to approximate the fixed points of Equation (2) even when the existence of fixed points is guaranteed.

To overcome this limitation, researchers in this direction developed different iterative processes to approximate fixed points of non-expansive mappings and other mappings that are more general than non-expansive mappings. We list some of the methods available in the literature. The Mann iterative [11] process is defined as follows:

$$x_{j+1} = (1 - \alpha_j)x_j + \alpha_j T(x_j), \ j \geq 0, \tag{4}$$

where $\{\alpha_j\}_0^\infty$ is a real sequence in $(0, 1]$. If $T$ is continuous and the Mann iterative scheme converges, then it converges to a fixed point of $T$. However, if $T$ is not continuous, then there is no assurance that the Mann process will converge to the fixed point of $T$. Kranselski [12] proposed a one-step fixed point iterative scheme denoted by (KM) as

$$x_{j+1} = \frac{x_j + T(x_j)}{2}, \ j \geq 0. \tag{5}$$

Kanwar et al. [13] proposed another one-parameter family of a one-step fixed point iterative scheme as follows:

$$x_{j+1} = \frac{mx_j + T(x_j)}{(m+1)}, \ m \geq 0, \ j \geq 0. \tag{6}$$

Picard, Mann and Kranselski iteration schemes are obtained from (6) by using different values of the parameter used. There are two-step fixed point iterative schemes available in the literature. Some of them are listed here. Ishikawa [14] proposed the following two-step iteration algorithm as

$$\begin{aligned} x_{j+1} &= (1 - \beta_j)x_j + \beta_j T(y_j), \\ y_j &= (1 - \alpha_j)x_j + \alpha_j T(x_j), \ j \geq 0, \end{aligned} \tag{7}$$

where $\{\alpha_j\}_0^\infty$ and $\{\beta_j\}_0^\infty$ are real sequences in $(0, 1]$. The Ishikawa iteration procedure is a generalization of Mann iteration, but there is no dependence between the convergence results of Mann and Ishikawa iterative procedures. Moreover, Ishikawa iteration is a two-step Mann iteration with two different parameter sequences. For $\beta_j = 0$, the Ishikawa iterative scheme reduces to the Mann iterative scheme. Agarwal et al. [15] defined the S-iteration process as follows:

$$\begin{aligned} x_{j+1} &= (1 - \beta_j)T(x_j) + \beta_j T(y_j), \\ y_j &= (1 - \alpha_j)x_j + \alpha_j T(x_j), \ j \geq 0, \end{aligned} \tag{8}$$

where $\{\alpha_j\}_0^\infty$ and $\{\beta_j\}_0^\infty$ are real sequences in $(0, 1]$. Hussain et al. [16] provided an example to show that iterative scheme (8) is faster than Mann and Ishikawa iterative schemes for Zamfirescu operators. Thianwan [17] proposed another two-step iteration scheme as follows:

$$\begin{aligned} x_{j+1} &= (1 - \beta_j)y_j + \beta_j T(y_j), \\ y_j &= (1 - \alpha_j)x_j + \alpha_j T(x_j), \ j \geq 0, \end{aligned} \tag{9}$$

where $\{\alpha_j\}_0^\infty$ and $\{\beta_j\}_0^\infty$ are real sequences in $(0, 1]$. Yildirim et al. [18] proved the convergence of the Thianwan iterative scheme and its equivalence with Picard, Mann and Ishikawa iterative schemes for Zamfirescu operators. Khan [19] developed the Picard–Mann hybrid iterative process, which was faster than almost all two-step iterative schemes at that time. This is given as follows:

$$\begin{aligned} x_{j+1} &= T(y_j), \\ y_j &= (1 - \alpha_j)x_j + \alpha_j T(x_j), \ j \geq 0, \end{aligned} \tag{10}$$

where $\{\alpha_j\}_0^\infty$ is a real sequence in $(0, 1]$.

In the attempt to find faster iterative schemes, researchers moved towards three-step iterative processes. These iterative schemes are mostly compositions of Picard and Mann iterative schemes. Karakaya et al. [20] proposed the following algorithm. For each $x_0 \in D$, the sequence $\{x_j\}_0^\infty$ is defined by

$$
\begin{aligned}
x_{j+1} &= T(y_j), \\
y_j &= (1 - \alpha_j)z_j + \alpha_j T(z_j), \\
z_j &= T(x_j),\ j \geq 0.
\end{aligned}
\tag{11}
$$

Ullah and Arshad [21] introduced the following iteration process:

$$
\begin{aligned}
x_{j+1} &= T(y_j), \\
y_j &= T(z_j), \\
z_j &= (1 - \alpha_j)x_j + \alpha_j(Tx_j),\ j \geq 0.
\end{aligned}
\tag{12}
$$

Abbas et al. [22] proposed the following iteration algorithm: for each $x_0 \in D$, the sequence $\{x_j\}_0^\infty$ is defined by

$$
\begin{aligned}
x_{j+1} &= (1 - \alpha_j)y_j + \alpha_j T(y_j), \\
y_j &= T(z_j), \\
z_j &= T(x_j),\ j \geq 0,
\end{aligned}
\tag{13}
$$

where $\{\alpha_j\}_0^\infty$ is a sequence of real numbers in $(0, 1]$. They proved that this iterative scheme converges faster than most of the existing schemes in the literature. They also proved that this method is equivalent to iterative schemes given by (11) and (12).

Akutsah and Narain [23] proposed the following iterative scheme:

$$
\begin{aligned}
x_{j+1} &= T((1 - \alpha_j)y_j + \alpha_j T(y_j)), \\
y_j &= T(z_j), \\
z_j &= (1 - \beta_j)x_j + \beta_j T(x_j),\ j \geq 0.
\end{aligned}
\tag{14}
$$

They also discussed results about the strong convergence and stability of the proposed scheme. This iterative scheme is faster than iterative schemes (11), (12) and (13). Gürsoy et al. [24] defined the iteration scheme as follows:

$$
\begin{aligned}
x_{j+1} &= T(y_j), \\
y_j &= (1 - \alpha_j)T(x_j) + \alpha_j T(z_j), \\
z_j &= (1 - \beta_j)x_j + \beta_j T(x_j),\ j \geq 0.
\end{aligned}
\tag{15}
$$

Ullah and Arshad [25] developed an iterative process as

$$
\begin{aligned}
x_{j+1} &= T(y_j), \\
y_j &= T((1 - \alpha_j)x_j + \alpha_j T(z_j)), \\
z_j &= (1 - \beta_j)x_j + \beta_j T(x_j),\ j \geq 0.
\end{aligned}
\tag{16}
$$

Nawab et al. [26] defined the K-iteration process as

$$
\begin{aligned}
x_{j+1} &= T(y_j), \\
y_j &= T((1 - \alpha_j)T(x_j) + \alpha_j T(z_j)), \\
z_j &= (1 - \beta_j)x_j + \beta_j T(x_j),\ j \geq 0.
\end{aligned}
\tag{17}
$$

In general, two qualities, namely fastness and stability, play important roles for an iterative scheme to be preferred over the other iterative schemes. In this paper, we have proposed a new one-parameter class of a one-step fixed point iterative method, which is faster than many existing one-step methods. We also extended this method to two-step and three-step iterative schemes.

## 2. Preliminaries

In this section, we give some important results and definitions, which are used to prove the main results.

Berinde [27] introduced a class of operators on an arbitrary space $E$ satisfying the following condition:

$$||T(x) - T(y)|| \leq 2\delta(||x - T(x)|| + \delta||y - T(y)||) \quad \forall x, y \in E, \tag{18}$$

where $\delta = max\{a, \frac{b}{1-b}, \frac{c}{c-1}\}$, $a \in (0,1)$ and $b, c \in (0, \frac{1}{2})$.

Osilike [28] considered a more general class than Berinde, which is given as follows: there exists $L \geq 0, \delta \in [0,1)$ such that for each $x, y \in E$,

$$||T(x) - T(y)|| \leq L(||x - T(x)||) + \delta||x - y||. \tag{19}$$

Imoru and Olatinwo [29] further extended the class of mappings of Berinde and Osilike using the following contractive condition: there exists $\delta \in [0, 1)$ and a monotonically increasing and continuous function $\phi : [0, \infty) \to [0, \infty)$ such that $\phi(0) = 0$, and for all $x, y \in E$, satisfies the condition

$$||T(x) - T(y)|| \leq \phi(||x - T(x)||) + \delta||x - y||. \tag{20}$$

**Lemma 1** ([9]). *If $\delta$ is a real number such that $0 \leq \delta < 1$ and $\{\epsilon_j\}$ is a sequence of positive numbers such that $\lim_{j\to\infty} \{\epsilon_j\} = 0$, then for any sequence of positive numbers $\epsilon_j$ satisfying*

$$u_{j+1} \leq \delta u_j + \epsilon_j, \quad j = 0, 1, 2 \ldots, \tag{21}$$

*we have $\lim_{j\to\infty} u_j = 0$.*

**Definition 1.** *Let $\{t_j\}$ be any arbitrary sequence in $\mathbb{R}$. Then, an iteration procedure $x_{j+1} = f(T, x_j)$, converging to fixed point $r$, is said to be T-stable if for $\epsilon_j = ||t_{j+1} - f(T, t_j)||, \forall j \in \mathbb{N}$, we have $\lim_{j\to\infty} \epsilon_j = 0$ if and only if $\lim_{j\to\infty} t_j = r$.*

**Definition 2** ([30]). *Suppose that $\{a_j\}$ and $\{b_j\}$ are two real sequences with limits $a$ and $b$, respectively. Then, $\{a_j\}$ is said to converge faster than $\{b_j\}$, if $\lim_{j\to\infty} \left| \frac{a_j - a}{b_j - b} \right| = 0$.*

## 3. Development of Method

Assume that Equation (2) has a fixed point $x = r$. Let

$$y = T(x) \tag{22}$$

represent the graph of the function $T(x)$. Let $x_0 \neq 0$ be an initial guess of the required fixed point and $T(x_0)$ be the corresponding point on the graph of the function $y = T(x)$ (Figure 1).

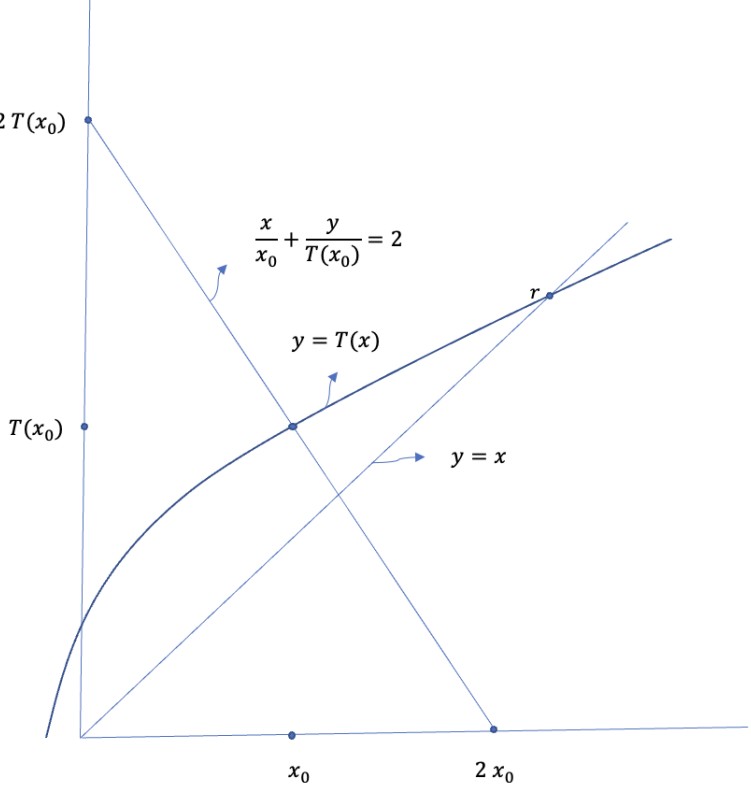

**Figure 1.** Graph of approximation of nonlinear function $y = T(x)$ by a linear approximation.

Here, we approximate the nonlinear function $y = T(x)$ by a linear approximation of the double intercept form of a straight line. Let $(x_0, T(x_0))$ be the mid point of a line segment between the axes. Then, the equation of the line is given by

$$\frac{x}{x_0} + \frac{y}{T(x_0)} = 2, \tag{23}$$

where $x_0, T(x_0) \neq 0$.

The point of intersection of line (23) and the line $y = x$ will give the required fixed point. Let $x = x_1$ be the point of intersection. Thus,

$$x_1 = \frac{2x_0 T(x_0)}{x_0 + T(x_0)}. \tag{24}$$

This can be generalized as

$$x_{j+1} = \frac{2x_j T(x_j)}{x_j + T(x_j)}, \quad j \geq 0. \tag{25}$$

We also propose the generalization of this iterative scheme as follows:

$$x_{j+1} = \frac{(k+1)x_j T(x_j)}{x_j + kT(x_j)}, \quad k \geq 0, \ j \geq 0. \tag{26}$$

or

$$x_{j+1} = \frac{(k+1)x_j T(x_j)}{kx_j + T(x_j)}, \quad k \geq 1, \ j \geq 0. \tag{27}$$

*3.1. Special Cases*

Here, we consider some special cases of expression (26).

1. For $k = 0$, formula (26) reduces to the simple Picard's fixed point method $x_{j+1} = T(x_j)$.

2. For $k = 1$, formula (26) gives rise to the harmonic mean formula $x_{j+1} = \frac{2x_j T(x_j)}{x_j + T(x_j)}$.

3. For $k = \frac{1-\alpha}{\alpha}$, where $\alpha \in (0, 1]$, formula (26) reduces to the following iterative scheme:

$$x_{j+1} = \frac{x_j T(x_j)}{\alpha x_j + (1-\alpha)T(x_j)}.$$

4. For $k = \frac{1-\alpha_j}{\alpha_j}$, where $\alpha_j$ is a real sequence $\in (0, 1]$, formula (26) corresponds to the following iterative scheme:

$$x_{j+1} = \frac{x_j T(x_j)}{\alpha_j x_j + (1-\alpha_j)T(x_j)}.$$

This is another variant of the Mann iterative scheme.

*3.2. Role of Parameter k*

In this section, we discuss the characteristics of parameter $k$.

Let $h(x) = \frac{(k+1)xT(x)}{x+kT(x)}$ and $h'(x) = \frac{(k+1)(x^2 T'(x)+k(T(x))^2)}{(x+kT(x))^2}$. Since $r$ is a fixed point of $T$, one gets

$$h'(r) = \frac{T'(r) + k}{(k+1)}.$$

For large values of $j$, $x_j \approx r$, one obtains

$$h'(x_j) = \frac{T'(x_j) + k}{(k+1)}.$$

Since $|h'(x_j)| < 1$ is a sufficient condition for the convergence of the proposed fixed point iterative scheme (26), one thus gets

$$|\frac{T'(x_j) + k}{(k+1)}| < 1.$$

This further implies that

$$-(2k+1) < T'(x_j) < 1. \tag{28}$$

This is the interval of convergence for proposed fixed point iteration given by (26), which is wider than the interval of convergence of Picard's iteration. In particular, for $k = 1$ (harmonic mean), the interval of convergence becomes $-3 < T'(x_j) < 1$.

Therefore, the harmonic mean fixed point scheme (HM) has a wider interval of convergence than simple Picard's fixed point iteration.

**Remark 1.** *There can be several ways to choose $T(x)$, but the choice of the $T(x)$ should be such that the fixed point iteration scheme converges to its fixed point. We present some examples to discuss this in detail.*

**Example 1.** *We have considered the following example to compare the proposed fixed point method, named the harmonic mean formula with Picard's iterative scheme.*

$$f(x) = x^3 + x - 1 \tag{29}$$

*Consider the following two possible rearrangements of $f(x)$ as*

*(1) $x_{j+1} = T(x_j) = 1 - x^3$, $j = 0, 1, 2, \ldots$,*

*(2) $x_{j+1} = T(x_j) = \sqrt[3]{1-x}$, $j = 0, 1, 2, \ldots$,*

*As $r = 0.6823278038$ is the required root of Equation (29) and the fixed point for the above two sequences with an initial guess $x_0 = 0.5$, in the case of $T(x) = 1 - x^3$, the Picard method diverges for the interval $[0, 1]$. Here, $T'(x) = -3x^2$ and $0 \leq x \leq 1$ implies that $-3 \leq -3x^2 \leq 0$. This further implies that $-3 \leq T'(x) \leq 0$, which violates the condition of $|T'(x)| < 1 \ \forall x \in [0, 1]$. As $T'(x) \in [-3, 1]$, the harmonic mean formula converges to the required fixed point.*

*In the case of $T(x) = \sqrt[3]{1-x}$, then $T'(x) = \dfrac{-1}{3(1-x)^{\frac{2}{3}}}$, which clearly implies that $|T'(x)| < 1$ $\forall x \in [0, 1]$, Picard's iterative method converges.*

**Example 2.** *Let us consider a square root finding problem by fixed point methods. Consider the function*

$$f(x) = x^2 - 9 \tag{30}$$

*Here, we consider two possible rearrangements of $f(x)$ as*

*(1) $x_{j+1} = T(x_j) = \dfrac{9}{x_j}$, $j = 0, 1, 2, \ldots$,*

*(2) $x_{j+1} = T(x_j) = \dfrac{3}{2}\left(\dfrac{x_j}{3} + \dfrac{3}{x_j}\right)$, $j = 0, 1, 2, \ldots$,*

*Here, $r = 3$ is the required root of Equation (30), and the fixed point for the above two sequences with an initial guess $x_0 = 2.9$. In the case of first sequence $|T'(x)| > 1$, Picard's method diverges, and the second one converges as $|T'(x)| < 1, \forall x \in [2.8, 3.05]$.*

*In the case of the first sequence, the corresponding harmonic mean iteration converges. In the case of the harmonic mean formula, the interval of convergence is $[-3, 1]$. In the case of the second sequence, the harmonic mean fixed point iterative scheme converges, as $|T'(x)| < 1, \forall x \in [2.8, 3.05]$.*

### 3.3. Two-Step and Three-Step Iterative Schemes

In this section, we propose new two-step and three-step iterative schemes defined as follows.

#### 3.3.1. Two-Step Iterative Schemes

For $x_0 \neq 0$ given in $D$, the sequence $\{x_j\}$ in $D$ is given by

$$
\begin{aligned}
x_{j+1} &= T\left(\frac{T(x_j) + y_j}{2}\right), \\
y_j &= \frac{(k+1)x_j T(x_j)}{x_j + kT(x_j)}, \quad j \geq 0, \ k \geq 0.
\end{aligned}
\tag{31}
$$

and

$$
\begin{aligned}
x_{j+1} &= T\left(\frac{x_j + y_j}{2}\right), \\
y_j &= \frac{(k+1)x_j T(x_j)}{x_j + kT(x_j)}, \quad j \geq 0, \ k \geq 0.
\end{aligned}
\tag{32}
$$

#### 3.3.2. Three-Step Iterative Scheme

For $x_0 \neq 0$ given in $D$, the sequence $\{x_j\}$ in $D$ is given by

$$
\begin{aligned}
x_{j+1} &= Ty_j, \\
y_j &= Tz_j, \\
z_j &= \frac{(k+1)x_j T(x_j)}{x_j + kT(x_j)}, \quad j \geq 0, \ k \geq 0.
\end{aligned}
\tag{33}
$$

## 4. Convergence and Stability Analysis

In this section, we shall prove the strong convergence and stability results and obtain error equations for iterative schemes given by (26), (31) and (33), respectively.

**Theorem 1.** *Let $T : D \to D$ be a function satisfying the contractive mapping (20) with fixed point 'r', where D is a non-empty closed and convex subset of real Banach space E. Let $\{x_j\}$ be defined by the iteration process (26) and $x_0 \in D$, where $k \geq 0$ is a real number. Then, $\{x_j\}$ converges strongly to a unique fixed point r of T.*

**Proof.** We shall establish that $\lim\limits_{j \to \infty}\{x_j\} = r$. Using (26), one has

$$
\begin{aligned}
||x_{j+1} - r|| &= ||\frac{(k+1)x_j T(x_j)}{x_j + kT(x_j)} - r|| \\
&= ||\frac{(k+1)x_j T(x_j)}{x_j + kT(x_j)} - \frac{x_j + kT(x_j)}{x_j + kT(x_j)} r|| \\
&\leq |\frac{kT(x_j)}{x_j + kT(x_j)}|\, ||x_j - r|| + |\frac{x_j}{x_j + kT(x_j)}|\, ||T(x_j) - r|| \\
&\leq |\frac{kT(x_j)}{x_j + kT(x_j)}|\, ||x_j - r|| + |\frac{x_j}{x_j + kT(x_j)}|\, (\phi(||r - Tr||) + \delta\, ||r - x_j||) \\
&= |\frac{kT(x_j)}{x_j + kT(x_j)}|\, ||x_j - r|| + |\frac{x_j}{x_j + kT(x_j)}|\, \delta\, ||r - x_j|| \\
&= \frac{|kT(x_j)| + |x_j\delta|}{|kT(x_j) + x_j|}\, ||x_j - r||.
\end{aligned}
\tag{34}
$$

Let $\frac{|kT(x_j)| + |x_j\delta|}{|kT(x_j) + x_j|} = \lambda_j$. Since $x_j$ and $T(x_j)$ have same sign and $k \geq 0, 0 < \delta < 1$, $x_j$ and $kT(x_j)$ also have the same sign. Thus, $|x_j + kT(x_j)| > |kT(x_j)| + |x_j\delta|$. Therefore,

$$
\lambda_j = \frac{|kT(x_j)| + |x_j\delta|}{|x_j + kT(x_j)|} < 1.
\tag{35}
$$

So, using (35) in (34), one gets

$$
||x_{j+1} - r|| \leq \lambda_j ||x_j - r||,
\tag{36}
$$

Repeating the above process, one gets

$$
\begin{aligned}
||x_{j+1} - r|| &\leq \lambda_j \cdot \lambda_{j-1} ||x_{j-1} - r|| \\
&\leq \lambda_j \cdot \lambda_{j-1} \cdot \lambda_{j-2} ||x_{j-2} - r||
\end{aligned}
$$

Continuing like this, one can write

$$
\begin{aligned}
||x_{j+1} - r|| &\leq \lambda_j \cdot \lambda_{j-1} \cdot \lambda_{j-2} \ldots\ldots \lambda_1 \cdot \lambda_0 ||x_0 - r|| \\
||x_{j+1} - r|| &\leq \prod_{n=0}^{j} \lambda_n ||x_0 - r||.
\end{aligned}
\tag{37}
$$

Let $\lambda = \max\limits_{0 \leq n \leq j}\{\lambda_n\}$. As each $\lambda_j < 1$, therefore $\lambda < 1$. Thus, (37) becomes

$$
||x_{j+1} - r|| \leq (\lambda)^j ||x_0 - r||.
\tag{38}
$$

Since $\lambda_j < 1$, $(\lambda)^j \to 0$ as $j \to \infty$. Hence, it follows from Lemma 1 that $\lim_{j \to \infty} ||x_{j+1} - r|| = 0$. Therefore, $\{x_j\}$ converges strongly to $r$.

We now prove that $r$ is unique. Let $r, r^*$ such that $Tr = Tr^* = r$. Suppose that $r \neq r^*$.

$$
\begin{aligned}
||r - r^*|| &= ||Tr - Tr^*|| \\
||r - r^*|| &\leq \phi(||r - Tr||) + \delta(||r - r^*||) \qquad \text{by using (20)} \\
&= \delta(||r - r^*||) \\
&< (||r - r^*||), \text{ as } \delta < 1.
\end{aligned}
\tag{39}
$$

This is a contradiction. Therefore $r = r^*$. $\square$

**Theorem 2.** *Let $T : D \to D$ be a function satisfying contractive condition (20) with fixed point $r$, where $D$ is a non-empty closed and convex subset of real Banach space $E$. Let $\{x_j\}$ be defined by the iteration process (26) and $x_0 \in D$, where $k \geq 0$ is a real number. Then, the iterative scheme is $T$-stable.*

**Proof.** Let $\{t_j\}$ be an arbitrary sequence in $D$ and suppose that the sequence generated by (26) is $x_{j+1} = f(T, x_i)$, which converges to the unique fixed point $r$ and that $\epsilon_j = ||t_{j+1} - f(T, x_j)||, \forall j \in \mathbb{N}$. To show that the iterative scheme is $T$-stable, we have to show that $\lim\limits_{j \to \infty} \epsilon_j = 0$ if and only if $\lim\limits_{j \to \infty} t_j = r$. Suppose that $\lim\limits_{j \to \infty} \epsilon_j = 0$. We find that

$$
\begin{aligned}
||t_{j+1} - r|| &= ||t_{j+1} - f(T, t_j) + f(T, t_j) - r|| \\
&\leq ||t_{j+1} - f(T, t_j)|| + ||f(T, t_j) - r|| \\
&= \epsilon_j + ||f(T, t_j) - r|| \\
&= \epsilon_j + ||\frac{(k+1)t_j T(t_j)}{t_j + kT(t_j)} - r|| \\
&= \epsilon_j + ||\frac{(k+1)t_j T(t_j)}{t_j + kT(t_j)} - \frac{t_j + kT(t_j)}{t_j + T(t_j)} r|| \\
&\leq \epsilon_j + \left|\frac{kT(t_j)}{t_j + kT(t_j)}\right| ||t_j - r|| + \left|\frac{t_j}{t_j + kT(t_j)}\right| ||T(t_j) - r|| \\
&\leq \epsilon_j + \left|\frac{kT(t_j)}{t_j + kT(t_j)}\right| ||t_j - r|| + \left|\frac{t_j}{t_j + kT(t_j)}\right| (\phi(||r - T(r)||) + \delta \, ||r - t_j||) \\
&= \epsilon_j + \left|\frac{kTt_j}{x_j + kTt_j}\right| ||t_j - r|| + \left|\frac{t_j}{t_j + kTt_j}\right| \delta \, ||r - t_j|| \\
&= \epsilon_j + \frac{k|T(t_j)| + |t_j \delta|}{|kT(t_j) + t_j|} (||t_j - r||).
\end{aligned}
$$

Since from (35), $\lambda_j = \frac{|kT(t_j)| + |t_j \delta|}{|t_j + kT(t_j)|} < 1$ and $\lim\limits_{j \to \infty} \epsilon_j = 0$, by using Lemma 1, one obtains $\lim\limits_{j \to \infty} t_j = r$.

Conversely, suppose that $\lim\limits_{j \to \infty} t_j = r$. Proceeding similarly to before, we have

$$
\begin{aligned}
\epsilon_j &= ||t_{j+1} - f(T, t_j)|| \\
&= ||t_{j+1} - r + r - f(T, t_j)|| \\
&\leq ||t_{j+1} - r|| + ||f(T, t_j) - r|| \\
&\leq ||t_{j+1} - r|| + \frac{|Tt_j| + |kt_j \delta|}{|kT(t_j) + t_j|} ||t_j - r||.
\end{aligned}
$$

Using the hypothesis $\lim\limits_{j \to \infty} t_j = r$, one gets $\lim\limits_{j \to \infty} \epsilon_j = 0$.

Hence, iteration process (26) is $T$-stable. $\square$

**Theorem 3.** *Let* $T : D \rightarrow D$ *be a function satisfying* (20) *with fixed point r, where D us a non-empty closed and convex subset of real Banach space E. Let* $\{x_j\}$ *be defined by the iteration process* (26) *and* $x_0 \in D$, *where* $k \geq 0$ *is a real number. Then, the iterative scheme has a linear order of convergence, and for* $k = -T'(r)$, *the order of convergence of the iterative scheme is two.*

**Proof.** Let

$$h(x) = \frac{(k+1) \, x \, T(x)}{k \, T(x) + x}. \tag{40}$$

Suppose $x_j \approx r$. Expanding $h(x_j)$ about $r$ by Taylor series expansion, one gets $h(x_j) = h(r) + h'(r)(x_j - r) + h''(r)\frac{(x_j-r)^2}{2!} + O((x_j - r)^3)$.

Therefore, one has

$$x_{j+1} - r = h'(r)(x_j - r) + h''(r)\frac{(x_j - r)^2}{2!} + O((x_j - r)^3). \tag{41}$$

As $h'(r) = \frac{k+T'(r)}{k+1}$ and $h''(r) = \frac{r^3 T''(r)(k+1)+2kr^2 T'(r)(k+1)-2k(T'(r))^2-2kr^2}{r^3(k+1)^2}$.

Substituting these values in (41), one obtains

$$e_{j+1} = \frac{k + T'(r)}{k+1}e_j + \frac{r^3 T''(r)(k+1) + 2kr^2 T'(r)(k+1) - 2k(T'(r))^2 - 2kr^2}{2r^3(k+1)^2}e_j^2 + O(e_j^3).$$

Therefore, scheme (26) has a linear order of convergence.

Furthermore, if $k = -T'(r)$, then

$$e_{j+1} = \frac{r^3 T''(r)(k+1) + 2kr^2 T'(r)(k+1) - 2k(T'(r))^2 - 2kr^2}{2r^3(k+1)^2}e_j^2 + O(e_j^3).$$

This implies that scheme (26) has at least a second order of convergence. □

**Theorem 4.** *Let* $T : D \rightarrow D$ *be a function satisfying contractive condition* (20) *with fixed point r, where D is a non-empty closed and convex subset of real Banach space E. Let* $\{x_j\}$ *be defined by the iteration process* (31) *and* $x_0 \in D$, *where* $k \geq 0$ *is a real number. Then,* $\{x_j\}$ *converges strongly to a unique fixed point r of T.*

**Proof.** We shall establish that $\lim_{j \to \infty}\{x_j\} = r$. Using (31), one has

$$
\begin{aligned}
||y_j - r|| &= ||\frac{(k+1)x_j T(x_j)}{x_j + kT(x_j)} - r|| \\
&= ||\frac{(k+1)x_j T(x_j)}{x_j + kT(x_j)} - \frac{x_j + kT(x_j)}{x_j + kT(x_j)}r|| \\
&\leq |\frac{kT(x_j)}{x_j + kT(x_j)}| \, ||x_j - r|| + |\frac{x_j}{x_j + kT(x_j)}| \, ||T(x_j) - r|| \\
&\leq |\frac{kT(x_j)}{x_j + kT(x_j)}| \, ||x_j - r|| + |\frac{x_j}{x_j + kT(x_j)}| \, (\phi(||r - T(r)||) + \delta \, ||r - x_j||) \\
&= |\frac{kT(x_j)}{x_j + kT(x_j)}| \, ||x_j - r|| + |\frac{x_j}{x_j + kT(x_j)}| \, \delta \, ||r - x_j|| \\
&= \frac{|kT(x_j)| + |x_j\delta|}{|kT(x_j) + x_j|} \, ||x_j - r||. 
\end{aligned}
\tag{42}
$$

Further, let $\frac{|kT(x_j)| + |x_j\delta|}{|kT(x_j) + x_j|} = \lambda_j$. Since $x_j$ and $T(x_j)$ have the same sign and $k > 0$, $0 < \delta < 1$, so $x_j$ and $kTx_j$ also have the same sign. Thus, $|x_j + kT(x_j)| > |kT(x_j)| + |x_j\delta|$

$$\lambda_j = \frac{|kT(x_j)| + |x_j\delta|}{|x_j + kT(x_j)|} < 1. \tag{43}$$

So, (42) becomes

$$||y_j - r|| \leq \lambda_j ||x_j - r||. \tag{44}$$

Now,

$$
\begin{aligned}
||x_{j+1} - r|| &= \left\| T\left( \frac{T(x_j) + y_j}{2} \right) - r \right\| \\
&= \left\| T\left( \frac{T(x_j) + y_j}{2} \right) - Tr \right\|, \\
&\leq \phi(||r - Tr||) + \delta \left\| r - \left( \frac{T(x_j) + y_j}{2} \right) \right\| \quad \text{by using (20)} \\
&= \frac{\delta}{2} ||2r - T(x_j) - y_j|| \\
&\leq \frac{\delta}{2} ||T(x_j) - r|| + \frac{\delta}{2} ||y_j - r|| \\
&\leq \frac{\delta}{2} (\delta + \lambda_j) ||x_j - r||. \tag{45}
\end{aligned}
$$

Repeating the above process, one gets

$$
\begin{aligned}
||x_{j+1} - r|| &\leq \frac{\delta}{2}(\delta + \lambda_j) ||x_j - r|| \\
||x_{j+1} - r|| &\leq \left( \frac{\delta}{2} \right)^2 (\delta + \lambda_j) \cdot (\delta + \lambda_{j-1}) ||x_{j-1} - r|| \\
&\leq \left( \frac{\delta}{2} \right)^3 (\delta + \lambda_j) \cdot (\delta + \lambda_{j-1}) \cdot (\delta + \lambda_{j-2}) ||x_{j-2} - r|| \\
||x_{j+1} - r|| &\leq \left( \frac{\delta}{2} \right)^{j+1} (\delta + \lambda_j) \cdot (\delta + \lambda_{j-1}) \cdot (\delta + \lambda_{j-2}) \dots \dots (\delta + \lambda_1) \cdot (\delta + \lambda_0) ||x_0 - r|| \\
||x_{j+1} - r|| &\leq \left( \frac{\delta}{2} \right)^{(j+1)} \left( \prod_{n=0}^{j} (\delta + \lambda_n) \right) ||x_0 - r||. \tag{46}
\end{aligned}
$$

Let $\lambda = \max_{0 \leq n \leq j} \{\lambda_n\}$. Since each $\lambda_j < 1$, so $\lambda < 1$. Then, (46) becomes

$$||x_{j+1} - r|| \leq \left( \frac{\delta}{2} \right)^{(j+1)} (\delta + \lambda)^{(j+1)} ||x_0 - r||. \tag{47}$$

Since $\delta < 1$ and $\lambda < 1$, one gets
$$\left( \frac{\delta}{2} \right)^{(j+1)} (\delta + \lambda)^{(j+1)} \to 0 \text{ as } j \to \infty.$$
Hence, it follows from (47) that $\lim_{j \to \infty} ||x_{j+1} - r|| = 0$. Therefore, $x_j$ converges strongly to $r$.

Similarly, as before, it can be easily checked that $r$ is unique. $\square$

**Theorem 5.** *Let $T : D \to D$ be a function satisfying contractive condition (20) with fixed point $r$, where $D$ is a non-empty closed and convex subset of real Banach space $E$. Let $\{x_j\}$ be defined by the iteration process (31) and $x_0 \in D$, where $k \geq 0$ is a real number. Then, the iterative scheme is T-stable.*

**Proof.** Let $\{t_j\}$ be an arbitrary sequence in $D$, and suppose that the sequence generated by (31) is $x_{j+1} = f(T, t_j)$ converging to unique fixed point $r$ and that $\epsilon_j = ||t_{j+1} - f(T, t_j)||, \forall j \in \mathbb{N}$. To show that the iterative scheme is $T$-stable, we have to show that $\lim_{j \to \infty} \epsilon_j = 0$ if and only if $\lim_{j \to \infty} t_j = r$. Suppose that $\lim_{j \to \infty} \epsilon_j = 0$. We have

$$
\begin{aligned}
||t_{j+1} - r|| &= ||t_{j+1} - f(T, t_j) + f(T, t_j) - r|| \\
&\leq ||t_{j+1} - f(T, t_j)|| + ||f(T, t_j) - r|| \\
&= \epsilon_j + ||f(T, t_j) - r|| \\
&= \epsilon_j + \left|\left| T(r) - T\left( \frac{T(t_j) + \frac{(k+1)t_j T(t_j)}{t_j + kT(t_j)}}{2} \right) \right|\right| \\
&= \epsilon_j + \phi(||r - T(r)||) + \delta \left|\left| r - \left( \frac{T(t_j) + \frac{(k+1)t_j T(t_j)}{t_j + kT(t_j)}}{2} \right) \right|\right| \\
&\leq \epsilon_j + \frac{\delta}{|2(kT(t_j) + t_j)|} \left( (|kT(t_j) + t_j| + |t_j|) \, ||r - T(t_j)|| + k \, |T(t_j)| \, ||t_j - r|| \right) \\
&\leq \epsilon_j + \frac{\delta}{2} \left( \left( \delta + \frac{k|T(t_j)|}{|kT(t_j) + t_j|} \right) ||t_j - r|| + \frac{|t_j|}{|kTt_j + t_j|} (\phi(||r - Tr||) + \delta ||t_j - r||) \right) \\
&= \epsilon_j + \frac{\delta}{2} \left( \delta + \frac{k|T(t_j)| + |\delta t_j|}{|kT(t_j) + t_j|} \right) ||t_j - r||.
\end{aligned}
\tag{48}
$$

Since from (35) $\lambda_j = \frac{|kT(t_j)| + |t_j \delta|}{|t_j + kT(t_j)|} < 1$ and $\lim_{j \to \infty} \epsilon_j = 0$, by using Lemma 1, one obtains $\lim_{j \to \infty} t_j = r$.

Conversely, suppose that $\lim_{j \to \infty} t_j = r$. Proceeding similarly to before, one can have

$$
\begin{aligned}
\epsilon_j &= ||t_{j+1} - f(T, t_j)||, \\
&= ||t_{j+1} - r + r - f(T, t_j)|| \\
&\leq ||t_{j+1} - r|| + ||f(T, t_j) - r|| \\
&\leq ||t_{j+1} - r|| + \frac{\delta}{2} \left( \delta + \frac{k|T(t_j)| + |\delta t_j|}{|kT(t_j) + t_j|} \right) ||t_j - r||.
\end{aligned}
$$

Using the hypothesis $\lim_{j \to \infty} t_j = r$, one gets $\lim_{j \to \infty} \epsilon_j = 0$.

Hence, iteration process (31) is $T$-stable. $\square$

**Theorem 6.** *Let $T : D \to D$ be a function satisfying contractive condition (20) with fixed point $r$, where $D$ is a nonempty closed and convex subset of real Banach space $E$. Let $\{x_j\}$ be defined by the iteration process (31) and $x_0 \in D$, where $k \geq 0$ is a real number. Then, this iterative scheme has a linear order of convergence.*

**Proof.** For a given sequence $\{x_j\}$, we denote $e_{x_j} = x_j - r$. By Taylor's series expansion about $r$, one gets

$$
\begin{aligned}
T(x_j) &= T(r) + T'(r)(x_j - r) + T''(r) \frac{(x_j - r)^2}{2!} + O((x_j - r)^3), \\
&= r + T'(r)e_{x_j} + T''(r) \frac{(e_{x_j})^2}{2!} + O(e_{x_j}^3).
\end{aligned}
$$

Using this expansion, one can write

$$
\begin{aligned}
e_{y_j} &= y_j - r \\
&= \frac{(k+1)x_j T(x_j)}{x_j + kT(x_j)} - r \\
&= \frac{(k+1)(e_{x_j}+r)(r+T'(r)e_{x_j}+T''(r)\frac{(e_{x_j})^2}{2!}+O(e^3_{x_j}))}{(e_{x_j}+r)+k(r+T'(r)e_{x_j}+T''(r)\frac{(e_{x_j})^2}{2!}+O(e^3_{x_j}))} - r \\
&= (1-k+k^2)(k+T'(r))e_{x_j} + \frac{(1-k+k^2)((k+1)T'(r)+\frac{1}{2!}T''(r))+(k+T'(r))(2k-1)(1+kT'(r))}{r}e^2_{x_j} \\
&\quad + O(e^3_{x_j}).
\end{aligned}
$$

Let $z_j = \frac{T(x_j)+y_j}{2}$. Thus, $e_{z_j} = \frac{T(x_j)+y_j}{2} - r$. This implies that

$$
e_{z_j} = \frac{T'(r)e_{x_j} + \frac{T''(r)e^2_{x_j}}{2} + e_{y_j}}{2}
$$

In addition,

$$
T(z_j) = r + T'(r)e_{z_j} + T''(r)\frac{(e_{z_j})^2}{2!} + O(e^3_{z_j}).
$$

Thus, one can write

$$
\begin{aligned}
x_{j+1} - r &= T(z_j) - r \\
e_{x_{j+1}} &= T(z_j) - r \\
e_{x_{j+1}} &= T'(r)e_{z_j} + T''(r)\frac{(e_{z_j})^2}{2!} + O(e^3_{z_j}).
\end{aligned}
$$

Using the value of $e_{z_j}$ in the above equation, one can get

$$
e_{x_{j+1}} = T'(r)\left(\frac{T'(r)e_{x_j}+\frac{T''(r)e^2_{x_j}}{2}+e_{y_j}}{2}\right) + \frac{T''(r)}{2!}\left(\frac{T'(r)e_{x_j}+\frac{T''(r)e^2_{x_j}}{2}+e_{y_j}}{2}\right)^2 + O\left(\left(\frac{e_{x_j}+e_{y_j}}{2}\right)^3\right).
$$

$$
e_{x_{j+1}} = \left(\frac{(T'(r))^2}{2} + T'(r)\left(1-k+k^2\right)(k+T'(r))\right)e_{x_j} + O(e^2_{x_j}).
$$

The above equation represents the error equation for the iterative scheme given by (31). □

**Theorem 7.** *Let $T : D \to D$ be a function satisfying contractive condition (20) with fixed point $r$, where $D$ is a nonempty closed and convex subset of real Banach space $E$. Let $\{\alpha_j\}_0^\infty$ be a sequence in $(0,1]$ and $k \geq 0$. Given $u_0 = x_0 \in D$, consider the sequence $\{x_j\}$ and $\{u_j\}$ obtained through the iteration processes (31) and (10), respectively. Then, $\{x_j\}$ converges to $r$ faster than $\{u_j\}$.*

**Proof.** From (47) in Theorem 4, we get the following inequality:

$$
||x_{j+1} - r|| \leq \left(\frac{\delta}{2}\right)^{(j+1)} \left(\prod_{n=0}^{j}(\delta + \lambda_n)\right) ||x_0 - r||. \tag{49}
$$

Using a similar argument to that in Theorem 4, the iteration process (10) takes the form

$$
||x_{j+1} - r|| \leq (\delta)^{(j+1)} \left(\prod_{n=0}^{j}(1 - \alpha_n(1-\delta))\right) ||u_0 - r||. \tag{50}
$$

Let

$$a_j = \left(\frac{\delta}{2}\right)^{(j+1)} \left(\prod_{n=0}^{j} (\delta + \lambda_n)\right) ||x_0 - r||,$$

$$b_j = (\delta)^{(j+1)} \left(\prod_{n=0}^{j} (1 - \alpha_n(1 - \delta))\right) ||u_0 - r||,$$

$$(51)$$

and

$$\Phi_j = \frac{a_j}{b_j} = \frac{\left(\frac{\delta}{2}\right)^{(j+1)} \left(\prod_{n=0}^{j} (\delta + \lambda_n)\right) ||x_0 - r||}{(\delta)^{(j+1)} \left(\prod_{n=0}^{j} (1 - \alpha_n(1 - \delta))\right)}. \tag{52}$$

Using $u_0 = x_0$ in (52), we obtain

$$\Phi_j = \frac{a_j}{b_j} = \frac{\left(\prod_{n=0}^{j} \left(\frac{\delta + \lambda_n}{2}\right)\right)}{\left(\prod_{n=0}^{j} (1 - \alpha_n(1 - \delta))\right)}. \tag{53}$$

This can be also written as

$$\Phi_j = \frac{a_j}{b_j} = \prod_{n=0}^{j} \frac{\left(\frac{\delta + \lambda_n}{2}\right)}{(1 - \alpha_n(1 - \delta))}. \tag{54}$$

Since $\delta \in (0,1)$ so, $\alpha_j < \frac{1}{2(1-\delta)} + \frac{1}{2}$, we get $(1 - \alpha_n(1 - \delta)) > \frac{\delta + \lambda_n}{2}$, which implies that $\frac{\left(\frac{\delta + \lambda_n}{2}\right)}{(1 - \alpha_n(1 - \delta))} < 1$.

Therefore, $\lim\limits_{j \to \infty} \Phi_j = 0$. Using Definition 2, we conclude that $\{x_j\}$ converges to $r$ faster than $\{u_j\}$. $\square$

**Remark 2.** *Proceeding in a similar way, it can be shown that the new iteration scheme (31) converges faster than other two-step iterative schemes.*

**Remark 3.** *By using a similar argument, we can prove the strong convergence stability and fastness results for the two-step iterative scheme (32) and three-step fixed point iteration (33).*

**Remark 4.** *The error equation for three-step fixed point iteration (33) is given as*

$$\begin{aligned}
e_{x_{j+1}} = \quad & T'(r)(T'(r)((1 - k + k^2)(k + T'(r)))e_{x_j} + \frac{T'(r)}{r}((1 - k + k^2)((k + 1)T'(r) \\
& + \frac{1}{2!}T''(r)) + (k + T'(r))(2k - 1)(1 + kT'(r)) \\
& + \frac{1}{2!}T''(r)(1 - k + k^2)^2(k + T'(r))^2 \\
& + \frac{1}{2!}T''(r)(T'(r)^2((1 - k + k^2)^2(k + T('r))^2 e_{x_j}^2 + O(e_{x_j}^3).
\end{aligned}$$

*Furthermore, if $k = -T'(r)$, then $e_{x_{j+1}} = \left(\frac{T'(r)}{r}((1 - k + k^2)(k + 1)T'(r) + \frac{1}{2!}T''(r)\right)e_{x_j}^2$ $+O(e_{x_j}^3)$, which implies that the iterative scheme (33) has second order convergence.*

## 5. Numerical Experiments

The theoretical results proposed in previous sections are tested in this section. We take some particular cases of our method (26) by substituting $k = \frac{1}{5}$ (*NI1*), $k = \frac{1}{20}$ (*NI2*), $k = \frac{1}{1000}$ (*NI3*) and $k = -T'(x_j)$ (*NI4*).

To check the effectiveness of the proposed method (26), we consider some scalar nonlinear equations, which are mentioned in Table 1. In addition, in Table 1, their corresponding fixed points and initial guesses are mentioned. For better comparison, in each example, we find the number of iterations required and the computational time of convergence to reach the stopping criteria given as follows:

$$|x_{j+1} - x_j| + |f(x_j)| < 10^{-15}. \tag{55}$$

It is well known that CPU time is not unique and depends on the specification of the computer. The mean CPU time is calculated by taking the mean of 12 performances of the program. The mean CPU time for each test problem in seconds is mentioned. The comparison results of different cases of (26) with Picard, Mann and KM are shown in Table 2. These results show that when $k$ tends to zero, the results are almost the same as for Picard iteration, and our proposed method performs much better than Mann and KM for each test problem in terms of the number of iterations required to attain the stopping criteria.

For the comparison of iterative schemes given by (31) and (33), we consider different scalar test problems in Table 3. In Table 4, we have compared different two-step iterative schemes given by Ishikawa (ISH), Agarwal (AGR), Thainwan (THI) and Khan (KHAN) with the proposed two-step scheme given by (31) called *NSI1* with $\alpha_j = \beta_j = \frac{1}{3+j^3}$ and $k = \frac{1}{20}$. Results show that our proposed two-step method performed much better than well-known similar existing two-step methods. In Tables 5 and 6, we have compared different three-step fixed point iterative schemes given by Karakaya, Ullah, Abbas, Akutash, Gürsoy and Nawab with the proposed three-step iterative scheme (33) and when $k = -T'(r)$, named *NTI1*, *NTI2*, respectively, with $\alpha_j = \beta_j = \frac{1}{3+j^3}$ and $k = \frac{1}{20}$.

Further, in Table 7, we consider the test problems of linear and non-linear systems of equations, and their results are shown in Tables 8 and 9. All these calculations have been performed in *Mathematica* 12 with multiple precision arithmetic using a laptop with an Apple MacBook Air M1 containing an Apple M1 chip, which has 8 Core CPU ($4 \times 3.2$ GHz and $4 \times 2.064$ GHz) with 8 GB of RAM and the MacOS Ventura (13.2.1) operating system.

**Table 1.** Test problems for one-step method.

| S.No. | Problem | Fixed Point Iteration | Root or Fixed Point $r$ | Initial Point $x_0$ |
|---|---|---|---|---|
| (1) | $f(x) = x^3 - 3x - 18$ | $T(x) = \sqrt[3]{3x + 18}$ | 3 | $x_0 = 1000$ |
| (2) | $f(x) = \sin(x) - 10(x - 1)$ | $T(x) = 1 + \frac{\sin(x)}{10}$ | 1.0885977523 | $x_0 = 1.1$ |
| (3) | $f(x) = x^2 - 3$ | $T(x) = \frac{3}{4}(x + \frac{1}{x})$ | 1.7320508075 | $x_0 = 3$ |
| (4) | $f(x) = 2x - \log_{10} x - 7$ | $T(x) = \frac{1}{2}(\log_{10} x + 7)$ | 3.7892782482 | $x_0 = 0.6$ |
| (5) | $f(x) = e^x - 3x$ | $T(x) = \frac{1}{3}e^x$ | 0.6190612867 | $x_0 = 1$ |
| (6) | $f(x) = \tan(x) - x$ | $T(x) = \pi + \tan^{-1}(x)$ | 4.4934094579 | $x_0 = 5$ |
| (7) | $f(x) = x - \frac{1}{2} - \sin(x)$ | $T(x) = \frac{1}{2} + \sin(x)$ | 1.4973003890 | $x_0 = 1.5$ |
| (8) | $f(x) = \cos x - xe^x$ | $T(x) = e^{-x} \cos x$ | 0.5177573636 | $x_0 = 0.52$ |
| (9) | $f(x) = x - \log(x + 2)$ | $T(x) = \log(x + 2)$ | 1.1461932206 | $x_0 = 1.5$ |
| (10) | $f(x) = x^3 + x^2 - 1$ | $T(x) = \frac{1}{\sqrt{x+1}}$ | 0.7548776663 | $x_0 = 0.5$ |

**Table 2.** Comparison of different fixed point methods on problems in Table 1.

| Problem | Number of Iterations | Picard | Mann | KM | NI1 | NI2 | NI3 | NI4 |
|---------|----------------------|--------|------|-----|------|------|------|------|
| | **CPU Time** | | | | | | | |
| (1) | NOI | 22 | * | 76 | **31** | **23** | **20** | **6** |
| | CPU | 3.71 | 3.78 | 3.73 | **3.72** | **3.73** | **3.68** | **3.70** |
| (2) | NOI | 11 | * | 50 | **21** | **14** | **11** | **3** |
| | CPU | 3.66 | 3.76 | 3.70 | **3.64** | **3.67** | **3.70** | **3.61** |
| (3) | NOI | 53 | * | 127 | **68** | **57** | **54** | **4** |
| | CPU | 3.62 | 3.67 | 3.66 | **3.64** | **3.61** | **3.67** | **3.63** |
| (4) | NOI | 14 | * | 58 | **25** | **17** | **14** | **6** |
| | CPU | 3.65 | 3.77 | 3.67 | **3.67** | **3.65** | **3.67** | **3.59** |
| (5) | NOI | 73 | * | 163 | **91** | **77** | **73** | **5** |
| | CPU | 3.48 | 3.56 | 3.46 | **3.54** | **3.51** | **3.47** | **3.44** |
| (6) | NOI | 13 | * | 57 | **24** | **16** | **13** | **4** |
| | CPU | 3.57 | 3.54 | 3.49 | **3.50** | **3.49** | **3.48** | **3.43** |
| (7) | NOI | 12 | * | 47 | **20** | **14** | **12** | **3** |
| | CPU | 3.41 | 3.46 | 3.45 | **3.48** | **3.46** | **3.44** | **3.43** |
| (8) | NOI | 195 | * | 13 | **45** | **29** | **14** | **12** |
| | CPU | 3.50 | 3.51 | 3.43 | **3.51** | **3.55** | **3.59** | **3.41** |
| (9) | NOI | 30 | * | 81 | **40** | **33** | **30** | **4** |
| | CPU | 3.53 | 3.48 | 3.46 | **3.46** | **3.52** | **3.57** | **3.42** |
| (10) | NOI | 23 | * | 37 | **9** | **19** | **23** | **5** |
| | CPU | 3.54 | 3.59 | 3.56 | **3.41** | **3.51** | **3.53** | **3.40** |

(*) Mann iteration does not attain stopping criteria up to 200 iterations.

**Table 3.** Test problems for two-step and three-step methods.

| S.No. | Problem | Fixed Point Iteration | Root or Fixed Point $r$ | Initial Point $x_0$ |
|-------|---------|------------------------|--------------------------|----------------------|
| (1) | $f(x) = x - \frac{1}{2} - \sin(x)$ | $T(x) = \frac{1}{2} + \sin(x)$ | 1.49730038909589 | $x_0 = 1.5$ |
| (2) | $f(x) = \cos x - xe^x$ | $T(x) = e^{-x} \cos x$ | 0.51775736368245 | $x_0 = 0.7$ |
| (3) | $f(x) = x - \log(x+2)$ | $T(x) = \log(x+2)$ | 1.14619322062058 | $x_0 = 1.5$ |
| (4) | $f(x) = x^3 - 3x - 18$ | $T(x) = \sqrt[3]{3x + 18}$ | 3 | $x_0 = 1000$ |
| (5) | $f(x) = \tan(x) - x$ | $T(x) = \pi + \tan^{-1}(x)$ | 4.49340945790906 | $x_0 = 4$ |
| (6) | $f(x) = 9x - 54$ | $T(x) = \sqrt{x^2 - 9x + 54}$ | 6 | $x_0 = 30$ |
| (7) | $f(x) = x^2 - 3$ | $T(x) = \frac{3}{4}(x + \frac{1}{x})$ | 6 | $x_0 = 1$ |

**Table 4.** Results of comparison of two-step methods on problems in Table 3.

| Problem | Number of Iterations | ISH | THI | AGA | KHAN | NSI1 |
|---------|----------------------|-----|-----|------|-------|-------|
| | **CPC Time** | | | | | |
| (1) | NOI | * | * | 12 | 11 | **6** |
| | CPU | 3.73 | 3.73 | 3.49 | 3.42 | **3.39** |
| (2) | NOI | * | * | 164 | 158 | **73** |
| | CPU | 4.03 | 4.07 | 4.11 | 3.96 | **3.83** |
| (3) | NOI | * | * | 30 | 29 | **16** |
| | CPU | 3.67 | 3.67 | 3.49 | 3.42 | **3.46** |

**Table 4.** *Cont.*

| Problem | Number of Iterations | ISH | THI | AGA | KHAN | NSI1 |
|---------|---------------------|-----|-----|-----|------|------|
| (4) | NOI | * | * | 20 | 19 | **10** |
|  | CPU | 3.45 | 3.49 | 3.41 | 3.43 | **3.39** |
| (5) | NOI | * | * | 12 | 11 | **7** |
|  | CPU | 3.79 | 3.80 | 3.49 | 3.39 | **3.46** |
| (6) | NOI | * | * | 35 | 34 | **18** |
|  | CPU | 3.48 | 3.44 | 3.3.43 | 3.42 | **3.36** |
| (7) | NOI | * | * | 51 | 51 | **26** |
|  | CPU | 3.46 | 3.49 | 3.44 | 3.39 | **3.38** |

(*) Ishikawa and Thiawan iterations do not attain stopping criteria up to 200 iterations.

**Table 5.** Results of comparison of three-step methods on problems in Table 3.

| Problem | Number of Iteration | Karakaya | Ullah 1 | Abbas | NTI1 | NIT2 |
|---------|---------------------|----------|---------|-------|------|------|
|  | **CPU Time** |  |  |  |  |  |
| (1) | NOI | 6 | 6 | 6 | **4** | **2** |
|  | CPU | 3.62 | 3.60 | 3.58 | **3.60** | **3.63** |
| (2) | NOI | 78 | 78 | 78 | **47** | **5** |
|  | CPU | 3.64 | 3.61 | 3.64 | **3.65** | **3.52** |
| (3) | NOI | 14 | 14 | 14 | **10** | **4** |
|  | CPU | 3.56 | 3.53 | 3.50 | **3.60** | **3.55** |
| (4) | NOI | 9 | 9 | 9 | **7** | **4** |
|  | CPU | 3.61 | 3.62 | 3.56 | **3.59** | **3.58** |
| (5) | NOI | 6 | 6 | 6 | **5** | **3** |
|  | CPU | 3.59 | 3.59 | 3.61 | **3.59** | **3.61** |
| (6) | NOI | 17 | 17 | 17 | **12** | **4** |
|  | CPU | 3.62 | 3.60 | 3.59 | **3.59** | **3.56** |
| (7) | NOI | 26 | 26 | 25 | **18** | **3** |
|  | CPU | 3.59 | 3.62 | 3.59 | **3.59** | **3.60** |

**Table 6.** Results of comparison of three-step methods on problems in Table 3.

| Problem | Number of Iterations | Akutash | Gürsoy | Ullah2 | Nawab | NTI1 | NTI2 |
|---------|---------------------|---------|--------|--------|-------|------|------|
|  | **CPU Time** |  |  |  |  |  |  |
| (1) | NOI | 6 | 6 | 5 | 4 | **4** | **2** |
|  | CPU | 3.58 | 3.60 | 3.59 | 3.60 | **3.60** | **3.63** |
| (2) | NOI | 74 | 82 | 79 | 54 | **47** | **5** |
|  | CPU | 3.67 | 3.64 | 3.67 | 3.65 | **3.65** | **3.52** |
| (3) | NOI | 14 | 15 | 15 | 10 | **10** | **4** |
|  | CPU | 3.52 | 3.56 | 3.55 | 3.52 | **3.60** | **3.55** |
| (4) | NOI | 10 | 10 | 9 | 7 | **7** | **4** |
|  | CPU | 3.58 | 3.63 | 3.63 | 3.61 | **3.59** | **3.58** |
| (5) | NOI | 6 | 7 | 6 | 5 | **5** | **3** |
|  | CPU | 3.56 | 3.57 | 3.63 | 3.61 | **3.59** | **3.61** |
| (6) | NOI | 16 | 17 | 16 | 12 | **12** | **4** |
|  | CPU | 3.61 | 3.60 | 3.63 | 3.59 | **3.59** | **3.56** |
| (7) | NOI | 25 | 25 | 26 | 17 | **18** | **3** |
|  | CPU | 3.58 | 3.59 | 3.59 | 3.58 | **3.59** | **3.60** |

**Table 7.** Test problem for linear and non-linear system of equations.

| S.No. | Problem | Fixed Point Iteration | Root or Fixed Point $r$ | Initial Point $x_0$ |
|---|---|---|---|---|
| (1) | $f_1(x,y,z) = 6x + y + z - 105$ <br> $f_2(x,y,z) = 4x + 8y + 3z - 155$ <br> $f_3(x,y,z) = 5x + 4y - 10z - 65$ | $T_1 = (105 - y - z)/6$ <br> $T_2 = (155 - 4x - 3y)/8$ <br> $T_3 = (65 - 5x - 4y)/-10$ | $(15, 10, 5)$ | $x_0 = (11, 12, 13)$ |
| (2) | $f_1(x,y) = \sin(xy) - x^2 - 5x - y$ <br> $f_2(x,y) = \cos(x + y) + y^2 - x - 6y + 2$ | $T_1 = (\sin(xy) - x^2 - y)/5$ <br> $T_2 = (\cos(x + y) + y^2 - x + 2)/6$ | * | $x_0 = \{1/10, 1/10\}$ |
| (3) | $f_1(x,y,z) = e^{-(x+z)} + 8z + 2y - 1$ <br> $f_2(x,y,z) = \cos(yz) + 3\sin(x) + 10x - 2z$ <br> $f_3(x,y,z) = 2y^2 + 10y - z + 2x + 2$ | $T_1 = -(\cos(yz) + 3\sin(x) - 2z)/10$ <br> $T_2 = -(2y^2 - z + 2x + 2)/10$ <br> $T_3 = -(e^{-(x+z)} + 2y - 1)/8$ | ** | $x_0 = \{\frac{-1}{10}, \frac{-1}{10}, \frac{-1}{10}\}$ |

(*) $r = \{-0.129497754736159, 0.558462490303788\}$, (**) $r = \{-0.0701895670135667, -0.188699903838307, 0.0438371350069641\}$.

**Table 8.** Results of comparison of two-step methods for system of equations.

| Problem | Number of Iterations | ISH | THI | AGA | KHAN | NSI1 |
|---|---|---|---|---|---|---|
| | **CPU Time** | | | | | |
| (1) | NOI | * | * | 54 | 52 | **27** |
| | CPU | 3.89 | 3.90 | 3.76 | 3.65 | **3.62** |
| (2) | NOI | * | * | 36 | 36 | **19** |
| | CPU | 5.34 | 5.42 | 4.02 | 3.84 | **3.78** |
| (3) | NOI | * | * | 27 | 26 | **14** |
| | CPU | 5.96 | 5.94 | 3.88 | 3.85 | **3.76** |

(*) Ishikawa and Thiawan iterations do not attain stopping criteria up to 200 iterations.

It can be observed that in most of the cases, new iterations perform better than most of the similar existing methods both in terms of the number of iterations and the computational time of convergence.

**Table 9.** Results of comparison of three-step methods for system of equations.

| Problem | Number of Iterations | Karakaya | Ullah1 | Abbas | Akutash | Gürsoy | Ullah2 | Nawab | NTI1 |
|---|---|---|---|---|---|---|---|---|---|
| | **CPU Time** | | | | | | | | |
| (1) | NOI | 27 | 27 | 27 | 26 | 27 | 27 | 19 | **18** |
| | CPU | 3.65 | 3.61 | 3.65 | 3.66 | 3.69 | 3.59 | 3.57 | **3.61** |
| (2) | NOI | 19 | 19 | 19 | 18 | 19 | 19 | 13 | **13** |
| | CPU | 3.98 | 4.02 | 4.23 | 4.53 | 4.01 | 4.12 | 4.06 | **4.11** |
| (3) | NOI | 14 | 14 | 14 | 13 | 14 | 14 | 10 | **9** |
| | CPU | 4.08 | 4.14 | 4.12 | 4.34 | 4.34 | 4.25 | 4.15 | **4.09** |

## 6. Conclusions

This paper presents a geometrically constructed family of one-parameter fixed point iterative schemes for approximating the fixed points of non-linear equations. The sufficient convergence criteria as well as the order of convergence are discussed in detail. We also extended this method to two-step and three-step fixed point iterative schemes. We have tested the proposed schemes on several numericals of scalar non-linear equations, systems of linear-equations and systems of non-linear equations and compared them with similar existing methods. The testing has been done on various parameters including the number of iterations and the CPU time required to attain stopping criteria. It was observed that the proposed schemes perform better than most of the existing schemes on both parameters.

**Author Contributions:** Conceptualization, A., M.N. and V.K.; Formal analysis, V.K.; Methodology, A., M.N. and V.K.; Writing—original draft, A.; Writing—review and editing, M.N. All authors have read and agreed to the published version of the manuscript.

**Funding:** Anku acknowledges the financial support of UGC, New Delhi, India. This research is funded by BININ01887801.

**Acknowledgments:** The authors would like to express their sincerest thanks to the editor and reviewers for their valuable suggestions, which significantly improved the quality of this paper.

**Conflicts of Interest:** The authors declare no conflict of interest.

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
