# Peer review of "On Some Fixed Point Iterative Schemes with Strong Convergence and Their Applications"

_mca, doi:10.3390/mca28020045_

Round 1
Reviewer 1 Report
The authors are mainly concerned with the problem of approximating fixed points of a contraction mapping on Banach space. They derived a new one-parameter class of fixed point iterative schemes geometrically. Further, they define new two-step and three-step iteration schemes from the proposed one-parameter class. In addition, they prove strong convergence, stability and carry out a comparative study with other well known iterative schemes available in literature. These results are new, interesting and mathematically correct. However, I propose the following minor revisions to increase the readability of this manuscript:
1. Initial guess should be different from zero (
).
2. Write down properly the coordinates of points on the graph in figure 1.
3. In equation (23) representing double intercept form: both
.
4. There is no need of the paragraph on page (5):
Since
implies that
, this further implies that
. This also implies that
. This is not correct mathematically
in general and is not used anywhere in the manuscript.
So exclude it from the manuscript.
Therefore, the manuscript can be accepted for publication, when they include these suggestions in the manuscript.

Author Response
Respected reviewer
Please find enclosed step-by-step reply to your comments through attachment file.
With regards
V. Kanwar

Reviewer 2 Report
In this paper, the authos propose two-step and
three-step fixed point iterative schemes for the the fixed points of contractive type mappings. The global convergence of the proposed methods is proved. Finally some numerical results demonstrate the effectiveness of the proposed methods.
However, some parts of this paper should be refined or detailed as
follows:
\\
{\bf Major problems:}
\begin{itemize}
\item[1)] It seems it is natural to specify and say which place takes offered method among known, to compare its to other methods, to accent its worth and shortages and to indicate a class of problem for which offered method is effective and better others.
\item [2)]
I suggest the authors to compare the proposed method with some well-known algorithms in the literature by providing the iteration numbers and the computational time.
To show the comparison of proposed method with other algorithms, one should use the definition of comparison and show it mathematically.
\end{itemize}
{\bf Minor Revisions:}
\begin{itemize}
\item[1)] Page 5, line 95 $x_{j+1}$ should be $x_{j+1}=\frac{x_jT(x_j)}{\alpha x_j+(1-\alpha)T(x_j)}.$
\item [2)] Page 5, line 102 $h(x)$ should be $h(x)=\frac{(k+1)xT(x)}{ x+kT(x)}.$
\item [3)] Page 8, after the third equality $\epsilon_j$ is missing.
\item [4)] Page 8, line 179 in (39) the second term the numerator should be $k|T(t_j)|+|\delta t_J|.$
\item [5)] Page 9, line 190 review the formula of $h''(r).$
\item [6)] Page 10, in (47) the second inequality should be as following
$$
\leq \frac{\delta}{2}(\frac{\delta}{2}\|x_{j-1}-r\|+\frac{\delta}{2}\lambda_{j-1}\|x_{j-1}-r\|)+\frac{\delta}{2}\lambda_j(\frac{\delta}{2}\|x_{j-1}-r\|+\frac{\delta}{2}\lambda_{j-1}\|x_{j-1}-r\|)
$$
\item [7)] A large number of papers by L-C. Ceng and his co-authors, Bnouhachem and his co-authors are missing in References.
\end{itemize}
To conclude, I believe that the presented work needs much stronger motivation before it could be considered for publication. Thus I cannot recommend the paper for publication in its present form.

Author Response

(The authors gave the same response as above.)

Round 2
Reviewer 2 Report
The paper is now improved a lot from the previous version. I am satisfied with the revisions and recommend the journal for accepting this manuscript.
Author Response
Dear reviewer
please find the attached file for your comment reply.
Regards
V. Kanwar
